# Lifestyle and Cardiometabolic Risk Factors Associated with Impoverishment Due to Out-of-Pocket Health Expenditure in São Paulo City, Brazil

**DOI:** 10.3390/ijerph21091250

**Published:** 2024-09-21

**Authors:** Lucas Akio Iza Trindade, Jaqueline Lopes Pereira, Jean Michel Rocha Sampaio Leite, Marcelo Macedo Rogero, Regina Mara Fisberg, Flavia Mori Sarti

**Affiliations:** 1School of Public Health, University of São Paulo, São Paulo 01246-904, Brazil; lucas.akio.trindade@usp.br (L.A.I.T.); jaque.lps@gmail.com (J.L.P.); jeanswb@usp.br (J.M.R.S.L.); mmrogero@usp.br (M.M.R.); rfisberg@usp.br (R.M.F.); 2School of Arts, Sciences and Humanities, University of São Paulo, São Paulo 03828-000, Brazil

**Keywords:** impoverishment, out-of-pocket expenditure, financial protection, obesity, cardiometabolic risk, physical activity

## Abstract

The rise in obesity and related chronic noncommunicable diseases (NCDs) during recent decades in Brazil has been associated with increases in the financial burden and risk of impoverishment due to out-of-pocket (OOP) health expenditure. Thus, this study investigated trends and predictors associated with impoverishment due to health expenditure, in the population of São Paulo city, Brazil, between 2003 and 2015. Household data from the São Paulo Health Survey (n = 5475) were used to estimate impoverishment linked to OOP health expenses, using the three thresholds of International Poverty Lines (IPLs) defined by the World Bank at 1.90, 3.20, and 5.50 dollars per capita per day purchasing power parity (PPP) in 2011. The results indicated a high incidence of impoverishment due to OOP disbursements for health care throughout the period, predominantly concentrated among low-income individuals. Lifestyle choices referring to leisure-time physical activity (OR = 0.766 at $3.20 IPL, and OR = 0.789 at $5.50 IPL) were linked to reduction in the risk for impoverishment due to OOP health expenditures whilst there were increases in the probability of impoverishment due to cardiometabolic risk factors referring to obesity (OR = 1.588 at $3.20 IPL, and OR = 1.633 at $5.50 IPL), and diagnosis of cardiovascular diseases (OR = 2.268 at $1.90 IPL, OR = 1.967 at $3.20 IPL, and OR = 1.936 at $5.50 IPL). Diagnosis of type 2 diabetes mellitus was associated with an increase in the probability of impoverishment at only the $1.90 IPL (OR = 2.506), whilst coefficients for high blood pressure presented lack of significance in the models. Health policies should focus on interventions for prevention of obesity to ensure the financial protection of the population in São Paulo city, Brazil, especially targeting modifiable lifestyle choices like promotion of physical activity and reduction of tobacco use.

## 1. Introduction

The United Nations’ agenda for the Sustainable Development Goals posits that monitoring financial protection in health represents a fundamental process for identification of advances towards universal healthcare coverage within national health systems worldwide. Financial protection of health refers to the guarantee of access to health care without incurring economic hardship at the household level, based on the incidence of catastrophic health expenditure and the corresponding risk of occurrence of impoverishment due to out-of-pocket (OOP) payments for health care [1,2].

Impoverishment due to OOP health expenses is characterized by a reduction in household income per capita to beneath subsistence levels, due to excessive OOP health disbursements, i.e., constraint of individuals’ daily income to levels lower than the International Poverty Line (IPL) thresholds in purchasing power parity (PPP), i.e., considering differences in living standards among countries. Consequently, a key element in the investigation of financial protection in national health systems refers to the choice of analytical approach for assessment of impoverishment, considering that its incidence depends on the proportion of household income compromised by health expenses in comparison to the IPL thresholds [3].

First, the incidence of impoverishment due to OOP health expenditures is influenced by the selection of the IPL threshold. Analyses of extreme impoverishment due to OOP health disbursements, based on the low IPL threshold ($1.90 IPL) indicate that it represents a major concern in developing countries. In addition, the incidence of extreme impoverishment has been decreasing globally during recent decades. In contrast, estimates of the incidence of impoverishment using higher IPL thresholds showed increasing trends in developed and developing countries between 2000 and 2010 [3].

Second, evidence from epidemiological studies usually lacks focus on financial protection in relation to individuals’ and families’ ability to pay, emphasizing absolute expenses related to health assistance within health systems [4,5]. Third, many studies investigating predictors linked to the occurrence of impoverishment due to OOP health expenditures have concentrated on the role of demographic and socioeconomic characteristics of populations in causing catastrophic OOP health expenditures; thus, the analysis of impoverishment has been considered as a secondary outcome, and other determinants of impoverishment have been neglected due to the scarcity of information on health conditions and lifestyle choices from surveys representative at population level [3,6,7].

Concerns regarding impoverishment due to OOP health expenses have been growing worldwide in recent decades due to the continuous increase in healthcare costs attributable to demographic and nutrition transitions (the rise in longevity and changes in population lifestyle) leading to epidemiological transition (decline in occurrence of morbimortality due to infectious diseases and growth in chronic noncommunicable diseases) [5,6,7,8,9]. Trends in population ageing, lifestyle, and health characteristics were consolidated in developed countries through the last decades of the 20th century and have been advancing in most developing countries from the turn of the 21st century onwards [8,9,10], including in Brazil [11].

The public health scenario in Brazil has been marked by progression of the epidemiological transition starting from the late 1980s; however, due to substantial disparities in demographic characteristics and socioeconomic conditions across regions, the national health system currently faces challenges imposed by the double burden of disease [11,12,13,14]. In particular, recent trends referring to the adoption of unhealthy diets, sedentary habits, and physical inactivity represent increased risk of cardiometabolic conditions, including obesity, cardiovascular diseases (CVDs), high blood pressure (HBP), and type 2 diabetes mellitus (DM), imposing substantial losses in quality of life, life expectancy, and financial resources for populations and national health systems worldwide [15,16,17].

Yet, many of the studies investigating financial protection in relation to health in Brazil lack evidence on the role of lifestyle and cardiometabolic risk factors on impoverishment due to OOP health expenditure [18,19,20]. Evidence relating to potential associations between nutritional status and impoverishment is scarce in the literature [21], and studies from other countries have generally reported weak associations between physical activity and financial protection in the context of health [22,23,24]. In contrast, findings from epidemiological studies in diverse countries indicate reductions in healthcare costs linked to adherence to physical activity recommendations, especially considering the effects of physical activity on body weight and the burden of noncommunicable diseases (NCDs) [25,26]. Thus, the present study investigated trends and predictors linked to the incidence of impoverishment due to out-of-pocket health expenditure in the population of São Paulo city, Brazil, between 2003 and 2015, particularly focusing the role of lifestyle and cardiometabolic risk factors, and referring to nutritional status, physical activity, tobacco use, and diagnosis of NCD.

## 2. Theoretical Framework

The study adopts the theoretical framework of Andersen’s Behavioral Model for Health Services Use for guidance of the analyses regarding the occurrence of impoverishment in São Paulo city, Brazil. Andersen’s model has been applied in empirical studies investigating predictors of the demand for health care, particularly assessing conditions related to the access and use of health care in diverse countries [27,28,29]. The foundations of Andersen’s model postulate the existence of a causal link between combinations of need, predisposition, and enabling factors at individual and contextual levels, in addition to the adoption of health behaviors, in the establishment of healthcare demand [30].

The conceptual structure of Andersen’s model defines the three types of determinants. Need factors are related to individuals’ health conditions, predisposing factors are linked to sociodemographic and behavioral characteristics of individuals, and enabling factors refer to the resources available to individuals that may facilitate their access to and use of health care. Therefore, Andersen’s model assumes that individuals’ ability to access and use health care comprises a multidimensional phenomenon, based on individuals’ characteristics and behaviors, contextual characteristics, and interactions within the health system. Thus, healthcare demand represents a dynamic situation that may be influenced by changes in modifiable elements corresponding to the conditions of individuals and their contexts [30].

The strengths of Andersen’s model rely on its holistic perspective on the access and use of healthcare within national health systems, presenting a flexible conceptual structure that may be adaptable to diverse situations in various national health systems worldwide [27,28,29]. Yet, the model has been criticized for lacking consideration of the qualitative dimensions of access to and use of health care due to its emphasis on quantitative dimensions; nevertheless, its major strength refers to the ability to generate robust analyses to support evidence-based decision making for public policy management, contributing to improvements in health systems worldwide [30].

Furthermore, the model has been previously adopted in studies examining financial protection within national health systems, especially referring to the occurrence of need factors that generate higher demand for health care, potentially resulting in catastrophic health expenditures [31,32]. In particular, Andersen’s model provides background for the investigation of financial protection in relation to healthcare, supporting the identification of predictors for inequalities influencing access to and use of healthcare [30].

The adaptation of Andersen’s Behavioral Model for Health Services Use in the context of the present study incorporates indicators at household level corresponding to the factors presented within the model framework (Figure 1):Need factors referring to health characteristics of household residents: presence of individuals with obesity, diagnosis of cardiovascular diseases, high blood pressure, and type 2 diabetes mellitus;Predisposing factors regarding sociodemographic characteristics of household residents: age, marital status, education attainment, occupational status, and residents in the household;Enabling factors encompassing economic characteristics of the household: income and private health insurance coverage.

In addition, personal health practices refer to tobacco use, physical activity level, and use of health services, comprising lifestyle characteristics and practices related to the use of health services and representing health behaviors intermediating the process of demand for health care.

In the present study, the application of the theoretical framework proposed in Andersen’s model allowed focus on attributes linked to the demand for health care associated with OOP health expenditure and the respective occurrence of impoverishment in São Paulo city, Brazil, under the following hypotheses:Impoverishment declined in São Paulo city between 2003 and 2015, despite the occurrence of demographic transition (ageing, predisposing factor), nutrition transition (increase in obesity, need factor), and epidemiologic transition (rise in chronic diseases, need factor), due to advances in the universal healthcare coverage provided by the public sector within the national health system;Out-of-pocket health expenditure and impoverishment were present in higher concentrations among lower-income households during the period, despite the advances in universal healthcare coverage provided by the public sector within the national health system;Changes in lifestyle choices related to health behaviors (tobacco use and physical activity) and need factors (obesity and chronic diseases) at population level between 2003 and 2015 had a substantial influence on the outcomes of the study (impoverishment and inequalities in OOP health expenditure).

## 3. Materials and Methods

### 3.1. Study Design

The present study focuses on quantitative analyses of data from an observational cross-sectional survey representative at population level conducted in São Paulo city, Brazil, the São Paulo Health Survey (ISA-Capital). São Paulo is the largest city in Brazil, encompassing approximately 12 million inhabitants. The aim of the ISA-Capital survey was to obtain data on demographic, socioeconomic, and lifestyle characteristics, health status, and healthcare utilization among individuals living in the urban area of the city. There are currently three editions of the ISA-Capital available for analysis, which were compiled by researchers from the School of Public Health at the University of São Paulo, Brazil, in collaboration with the Health Department of São Paulo city in 2003, 2008, and 2015. The survey sample was selected based on stratified probabilistic procedures, encompassing two stages: census tracts and households. Households were randomly selected within census tracts, and trained researchers applied structured questionnaires to individuals residing in the households. Further information on sampling procedures and research instruments have previously been published elsewhere [33,34].

The data collection procedures involved visits to the households selected in the sample, and invitations to residents to participate in the survey. Researchers trained to perform field work gathered information on individuals’ characteristics regarding demographic, socioeconomic, lifestyle, and health characteristics, in addition to household characteristics. The data collection tools comprised predominantly closed-ended questions within questionnaires previously tested in other surveys and validated for application in the Brazilian population.

### 3.2. Data

The dataset used in this study aggregated information from the participants of the three editions of the survey, using only variables that were directly comparable throughout the period of analysis. Missing cases and variables based on questions that had changed substantially in the different editions of the survey were excluded from this study to avoid potential bias in the statistical analyses. The organization of the survey data at individual level from 2003, 2008, and 2015 into a single dataset allowed the estimation of indicators referring to the occurrence of need factors (diagnosis of obesity and chronic diseases), predisposing factors (sociodemographic characteristics), and enabling factors (financial characteristics) at household level, in addition to the identification of health behaviors (health practices) and outcomes (out-of-pocket health expenditure and impoverishment) for the analyses.

### 3.3. Variables

#### 3.3.1. Outcome Variables

The outcome variables in this study corresponded to the occurrence of impoverishment and OOP health expenditure according to household income level. The occurrence of impoverishment was based on the three International Poverty Line (IPL) thresholds defined by the World Bank at 1.90, 3.20, and 5.50 dollars per capita per day purchasing power parity (PPP) in 2011 [1]. The analyses included the three thresholds of IPL for assessment of impoverishment, to allow capture of differences in income distribution and to minimize potential bias due to selection of only one IPL threshold [1], considering the substantial inequality in income distribution in Brazil [35].

The estimation of impoverishment was based on the thresholds of IPL established by the World Bank in 2017 (2011 PPP), considering that the data in the present study referred to the period 2003–2015 [36], in addition to the lack of national poverty lines officially adopted by the Brazilian government [37]. In fact, official documents from the Brazilian federal government and related research agencies (e.g., the Brazilian Institute for Geography and Statistics) have adopted the international poverty thresholds, considering their suitability for diverse country-level contexts [38,39].

Thus, the occurrence of impoverishment was estimated using standardized methodology, represented in Equations (1) and (2):(1)$$N_{j}=\frac{\sum W_{ij}-\sum H_{j}}{r_{j}}$$
(2)$$N_{j}>{IPL}_{n}\Rightarrow I_{jn}=0\;\vee \;N_{j}\leq {IPL}_{n}\Rightarrow I_{jn}=1$$
where *N_j_* = net income per capita per day of the household *j*; *W_ij_* = daily wage of the *i*^th^ individual in the household *j*; *H_j_* = daily health expenditure of residents in the household *j*; *r_j_* = residents in the household *j*; *IPL_n_* = International Poverty Line threshold *n*; *I_jn_* = occurrence of impoverishment (no = 0; yes = 1) in the household *j* at the threshold *n*.

The daily health expenditure of residents in the household corresponded to the mean disbursements per individual per day for medicines, private health insurance, appointments with physicians and other health professionals, dental treatments (including appointments with dentists and expenses for dental prosthesis), hospitalizations, medical equipment (including expenses relating to opticians, prosthetics, and orthotics), home nursing, and other expenses for health care.

The OOP health expenditure according to household income level represented the indicator of the financial burden due to disbursements related to healthcare demands in diverse socioeconomic strata, allowing the analysis to account for income-related inequality in access to and use of health care within the health system.

Variables expressed in monetary values were updated to December 2015 and converted into international monetary units in PPP to allow comparison throughout the period of analysis. The inflation rate adopted in the present study was the Brazilian Broad Consumer Price Index from the Brazilian Institute for Geography and Statistics (IPCA-IBGE), applying the 2011 PPP conversion factor from the World Bank; i.e., analyses were performed using monetary variables updated and transformed into international currency.

#### 3.3.2. Variables of Interest

The variables of interest in the study corresponded to the need factors and personal health practices of household residents, i.e., household-level indicators referring to the occurrence of lifestyle and cardiometabolic risk factors among residents (binary variables: no = 0; yes = 1): presence of residents with obesity, cardiovascular disease (CVD), high blood pressure (HBP), type 2 diabetes mellitus (DM), and health behaviors regarding use of tobacco and recommended levels of physical activity during leisure.

Bodyweight status of individuals was assessed through the estimation of Body Mass Index (BMI), based on weight (in kilograms) divided by squared height (in meters^2^), measures that were assessed during the application of the questionnaire in the households. Adult and elderly individuals with BMI equal or higher than 30.0 kg/m^2^ were considered obese, according to the recommendations of the World Health Organization [40]. Adolescents under 18 years old were considered obese according to the z-score of their BMI (z-score > 2), using the WHO Child Growth Standards 2007 package [41].

Variables related to the health status of individuals in the household were based on medical diagnosis for DM, HBP, and CVDs (including angina, coronary heart disease, arrhythmia, myocardial infarction, and other heart diseases). Use of tobacco was assessed through self-declaration of daily smoking habit.

The adherence to recommendations for physical activity level was defined through application of the long version of the International Physical Activity Questionnaire (IPAQ), translated into Portuguese and validated for the Brazilian population [42,43], converted into minutes of physical activity during leisure per week. Adolescents reporting ≥300 min of moderate or vigorous physical activity per week and adults or elderly individuals reporting ≥150 min of moderate or vigorous physical activity per week were considered to attain recommended physical activity levels [44].

#### 3.3.3. Control Variables

Additional variables were included in the analyses to allow control for the presence of predisposing and enabling factors (social, demographic, and financial characteristics of household residents), according to the framework of Andersen’s model. Table 1 summarizes the variables included in the analyses performed in the present study.

Demographic and socioeconomic characteristics: presence of elderly individuals (binary variable for presence of individuals ≥80 years old, to account for ageing) in the household; marital status (binary variable for married/accompanied in comparison to individuals without a companion); educational attainment (binary variable for higher education/college degree in relation to lower educational levels); occupational status of head of family (binary variable for employed); individuals living in the household; household income (binary variable for households in the upper tertile of income distribution);Health characteristics: private health insurance (PHI) coverage (binary variable for insurance ownership among household members); utilization of healthcare services (binary variables for occurrence of physician visits, hospitalizations, or dentist visits during the 12 months prior to the date of the survey);Survey characteristics: year of the ISA-Capital edition.

Household income per capita was estimated through the sum of personal earnings of household residents; these values were directly employed for analysis of inequality in OOP health expenditure. Tertiles of households’ income per capita were calculated and converted into the binary variable representing the upper tertile of income to allow capture of the effects of income level in the context of impoverishment, due to the disproportionate burden among lower-income households in Brazil. The educational attainment of the head of the household was declared according to categories of education and was converted into a binary variable to represent higher educational achievement in comparison to counterparts, to consider its influence on healthcare demand.

### 3.4. Statistical Analysis

The present study adopted two empirical strategies: the first strategy used a two-part model to investigate predictors associated with OOP health expenditure, and the second strategy assessed predictors linked to impoverishment due to OOP health expenditures at the three IPL thresholds through logistic models.

The two-part model encompassed the estimation of logistic regression for analysis of the probability of occurrence of outcomes greater than zero, and the generalized linear model—GLM (gamma family with log-link)—for analysis of associations between independent variables in relation to the outcome variable (OOP health expenditure). The adoption of two-part models accounts for excess of zeros in the investigation of healthcare expenses, considering that population surveys generally present zero-inflated distributions in healthcare demand [45].

Therefore, the two-part model improved the data fit, avoiding the interference of the excess of zeros in the estimates. Additionally, the strategy allowed estimation of the marginal effects of the coefficients obtained in the two parts of the model, identifying mean expected effects of the independent variables on the outcome variable (OOP health expenditure) [46]. Furthermore, the concentration of OOP health expenditure according to household income level per capita was estimated using the Kakwani and Gini indexes, showing trends in occurrence of health disbursements across socioeconomic groups [47].

The assessment of predictors linked with impoverishment due to OOP health expenditureswas based on estimation of odds ratios through logistic regression models for the three thresholds of IPL ($1.90, $ 3.20, and $ 5.50 dollars per capita per day PPP). The odds ratios for the variables of interest in relation to the outcome variable (impoverishment) allowed identification of the effects of lifestyle and cardiometabolic risk factors on the occurrence of impoverishment. The statistical analyses were conducted in Stata^®^ (StataCorp., College Station, TX, USA) version 18.0, using complex sample design to ensure representativeness at the population level, adjusting for potential correlations between subgroups of the sample, and adopting a statistical significance level of 5% (*p* < 0.05).

### 3.5. Ethical Aspects

The present study was approved by the Ethics Committee of the School of Public Health of the University of São Paulo, Brazil (CAAE 48271721.4.0000.5421). The three editions of the ISA-Capital survey were approved by the Ethics Committee of the School of Public Health (CAAE 003.0.162.000-08; 32344014.3.0000.5421; 36607614.5.0000.5421), and the Municipal Health Department of São Paulo city (CAAE 32344014.3.3001.0086). Individuals provided informed consent before participating in the survey, in accordance with the ethical principles of the Declaration of Helsinki.

## 4. Results

### 4.1. Sample Characteristics

A minority of households in the sample presented elderly individuals and heads of family with higher education. There was a significant increase in the proportion of households with heads of the family with higher education throughout the period, whilst there was a decrease in the proportion of households with more than four residents. A substantial number of households in São Paulo city were categorized in the high-income level across the period from 2003 to 2015 (Table 2).

Although there was low proportion of households with individuals with obesity, CVD, HBP, and DM in the sample, the proportion of households including individuals diagnosed with NCD increased throughout the period. Major part of the households presented lack of individuals using tobacco or practicing recommended level of physical activity during leisure, yet, trends showed significant change from 2003 to 2015 (Table 3).There was low coverage of private health insurance in the sample. The utilization of health services was concentrated on physician consultations, and the demand for health care (physician visits, hospitalizations, and dentist visits) increased throughout the period from 2003 to 2015 (Table 3).

### 4.2. Impoverishment Due to OOP Health Expenditures

The findings of the study suggest high incidence of and stable trends in impoverishment, and regressive inequality patterns in the burden of OOP health expenditure among households in São Paulo, Brazil, between 2003 and 2015. Approximately 2.5% of the households in the sample were driven to extreme poverty (IPL ≤ 1.90) due to OOP health expenditure; whilst 3.7% and 5.6% of the households were constrained according to the intermediate and high IPL thresholds, respectively (Table 4).

The stratification of the sample according to income levels showed higher incidence of impoverishment among low- and middle-income households compared with high-income households, excluding the trend of impoverishment among low- and middle-income level households according to the higher IPL (Table 4).

Furthermore, the estimation of the Kakwani and Gini indexes indicated concentration of the OOP health expenditure among low-income households, confirming the regressive nature of the financial burden represented by healthcare demand in the population of São Paulo city (Table 5).

Nevertheless, there were decreasing trends in the concentration indexes through the period of analysis, showing a gradual rise in household income and increase in financial protection within the Brazilian health system (Table 5).

### 4.3. Predictors Associated with OOP Health Expenditures and Impoverishment in Health

The estimation of the two-part model indicated that households with elderly individuals, those including residents diagnosed with CVD, and those categorized in the high-income level presented significant probability of incurring OOP health expenditure in addition to substantial positive effects in relation to the amount of health disbursement, similarly to households where head of the family was married or had completed higher education (Table 6).

Although households with individuals diagnosed with HBP presented a higher probability of incurring OOP health expenditure, the positive effect of disbursements in relation to healthcare lacked significance. The generalized linear model indicated that the presence of individuals fulfilling recommendations of physical activity during leisure was associated with significantly lower OOP health expenditure. In contrast, the presence of individuals with obesity or DM was associated with higher OOP expenditure (Table 6).

Sociodemographic predictors linked to higher probability of impoverishment due to OOP health expenditures included the presence of more than four residents in the household (OR = 1.813 at $ 1.90 IPL, and OR = 1.491 at $ 3.20 IPL). Predictors associated with lower probability of impoverishment were high income level (OR = 0.359 at $ 1.90 IPL, OR = 0.312 at $ 3.20 IPL, and OR = 0.252 at $ 5.50 IPL) and employment status of the head of the family (OR = 0.505 at $ 1.90 IPL, and OR = 0.564 at $ 3.20 IPL).

Private health insurance ownership was associated with a decline in risk of impoverishment at moderate and high IPL thresholds (OR = 0.639 at $ 3.20 IPL, and OR = 0.629 at $ 5.50 IPL). The utilization of health services, referring to hospitalization (OR = 1.705 at $ 3.20 IPL) and dentist visits (OR = 1.604 at $ 1.90 IPL, and OR = 1.828 at $ 5.50 IPL), increased the likelihood of impoverishment.

Regarding lifestyle factors, the presence of individuals achieving recommendations of physical activity during leisure (OR = 0.766 at $ 3.20 IPL, and OR = 0.789 at $ 5.50 IPL) decreased the probability of impoverishment, whereas the presence of individuals using tobacco products (OR = 1.689 at $ 1.90 IPL) increased the risk for extreme impoverishment.

Finally, odds of impoverishment increased due to the presence of residents with cardiometabolic risk factors, including obesity (OR = 1.588 at $ 3.20 IPL, and OR = 1.633 at $ 5.50 IPL) and diagnosis of cardiovascular disease (OR = 2.268 at $ 1.90 IPL, OR = 1.967 at $ 3.20 IPL, and OR = 1.936 at $ 5.50 IPL). Diagnosis of diabetes was associated with an increase in the probability of impoverishment only at the $ 1.90 IPL (OR = 2.506), whereas high blood pressure was not significant in the models (Table 7).

## 5. Discussion

The present study highlights the increase in occurrence of obesity and NCDs among individuals in Sao Paulo city from 2003 to 2015. The findings emphasize the ongoing processes of nutritional and epidemiological transitions in Brazil, which have presented substantial challenges to the national health system during recent decades [48]. The results of the present study further contribute to recognizing the impacts of changes in demography and lifestyle within the Brazilian population, especially regarding aspects of impoverishment at the household level.

In the present study, the incidence of impoverishment due to OOP health expenditure was approximately 2% at the extreme poverty line, 3% at the intermediate IPL (<3.20), and 5% at the high IPL (<5.50), being significantly concentrated among low-income households. These results corroborate previous evidence showing occurrence of approximately 4% impoverishment in Brazil [18]; however, our study contributes to the literature by pointing to the considerable inequalities in the burden of OOP health expenditure and impoverishment among lower-income households. That is, although there is high incidence of impoverishment due to catastrophic health expenditure in São Paulo city, similarly to other cities in Brazil and other Latin American and Caribbean countries during recent decades [7,49], the regressive nature of OOP health expenditure still represents a sizeable challenge within the Brazilian health system.

The current findings on the regressive incidence of OOP health expenditure are supported by previous evidence of inequalities in catastrophic health expenditure at the national level in Brazil [20] and Chile [50]. However, our results emphasize the financial burden imposed by OOP health expenditure causing impoverishment for households, showing the disproportional occurrence of impoverishment among lower-income individuals at the three IPL thresholds. Therefore, public policies directed towards financial protection relating to health care are associated with social determinants, requiring integrated strategies incorporating focus on dimensions of educational attainment, income distribution, and eradication of poverty [7].

In addition, this study identified significant effects of lifestyle and cardiometabolic risk factors on the occurrence of impoverishment, particularly in relation to physical activity, obesity, DM, and cardiovascular diseases. The role of physical activity for bodyweight management and health promotion has been emphasized in several studies, including its impacts on CVD, HBP, DM, and health expenditure [51,52,53,54]. Lack of physical activity has been linked to a substantial proportion of health expenditure among elderly individuals in Brazil, showing that part of the disbursements generally associated with obesity may be caused by changes in BMI due to physical inactivity [51]. A recent published review of the literature on the effects of physical activity showed robust evidence of impacts in relation to weight loss among individuals with overweight or obesity, and reduction in the risk of comorbidities [52], in addition to protective effects of physical activity in relation to catastrophic health expenditure among older adults in Brazil [19].

Furthermore, findings from studies focusing public sector expenditure attributable to physical inactivity, overweight, and obesity in the Brazilian health system identified considerable financial burden due to lifestyle and cardiometabolic risk factors [53,54], especially in relation to direct costs of hospitalizations associated with NCD [53] and obesity [51]. A recent study highlighted the persistent prevalence of sedentary behavior in the Brazilian population during recent decades [55]. Thus, the results of the present study present additional evidence for the private sector, identifying the burden of lifestyle and cardiometabolic risk factors in relation to impoverishment at the household level.

Previous studies conducted in other low- and middle-income countries support the results of the present study [22,56,57,58]. The presence of individuals with CVD in the household was risk factor for catastrophic health expenditure in South Africa and Ghana [22]. Households with underweight or overweight female heads of family presented higher risks of incurring catastrophic health expenditure in China [56], and there was higher risk of occurrence of impoverishment among households with residents diagnosed with NCDs, particularly CVDs, in Saudi Arabia [57] and in Serbia [58]. Therefore, the implementation of public policies focusing on promotion of healthy lifestyles, particularly maintenance of adequate body weight and adherence to the WHO recommendations for leisure-time physical activity, comprises an essential strategy for financial protection in relation to healthcare within populations [25].

In addition, public policies focusing promotion of equity and financial protection within national health systems should primarily target low- and middle-income households, considering their higher risk of impoverishment and the regressive nature of OOP health expenditure identified in the present study. Furthermore, evidence also shows occurrence of income-related inequalities in health behaviors comprising determinants of cardiometabolic health in the Brazilian population, including lower adherence to physical activity recommendations [59], higher occurrence of tobacco use [60], and lower diet quality [61,62].

Inequalities in OOP health expenditures remain a challenge in the context of the Brazilian health system, especially among households with residents diagnosed with NCDs or individuals with obesity [63]. Strategies to address obesity prevention and treatment at the primary healthcare level should acknowledge problems arising from the nutritional transition in Brazil, through a holistic approach to health based on promotion of healthy lifestyles [49]. Yet, a recent study showed that healthcare professionals within the public sector in the Brazilian health system (Unified Health System, SUS) considered that tackling obesity still represented a challenge due to lack of specialized training, inadequate professional communication, and the presence of comorbidities that impose limits on certain treatment protocols [64].

Despite progresses in the pursuit of universal health coverage based on the establishment of delivery of health care and distribution of medication free of charge in Brazil during recent decades [65,66], there are persistent inequalities in impoverishment. The two-tier design of the Brazilian health system ensures operationalization of assistance through the public and private sectors, the latter being financed through resources from charitable institutions, direct OOP disbursements, or private health insurance. Certain studies suggest that the demand for private healthcare in Brazil is linked to the perception of quality of care, especially referring to waiting times and access to technologies and specialized care. Therefore, individuals in Brazil may prefer to access private healthcare for highly complex or emergency treatment, incurring higher healthcare costs [67,68,69,70].

In the current study, the identification of financial protection against impoverishment due to OOP health expenditures through ownership of private health insurance at the intermediate (IPL < 3.20) and high (IPL < 5.50) thresholds reinforces evidence from previous research suggesting that households may identify economic incentives in private health insurance, particularly in terms of financial protection related to the need for curative care (e.g., inpatient services), and also that it may be associated with higher adherence to healthy lifestyles, including smoking cessation and regular practice of physical activity [70]. However, there are debates on the role of private health insurance in the context of the Brazilian health system, considering the evidence for weak association with avoidance of catastrophic health expenditure [71].

The study presents certain limitations. First, the use of cross-sectional data hinders the establishment of causal relations between covariates and the occurrence of impoverishment. However, considering the lack of representative data at the population level in Sao Paulo city, the present investigation comprises an original contribution to the field of knowledge relating to the evolution of and the predictors associated with impoverishment in the population of the largest city in Brazil, especially considering the availability of data on OOP health expenditure, lifestyle characteristics, and cardiometabolic risk factors throughout the period from 2003 to 2015.

Second, the use of self-reported economic information (household income and out-of-pocket healthcare expenditure), healthcare utilization (inpatient, outpatient, and dental care), and lifestyle characteristics (tobacco use) may have included errors in individuals’ declarations. Third, the survey lacks data on certain health behaviors that may be linked to OOP health expenditure and impoverishment, e.g., alcohol abuse, which has been significantly associated with impoverishment in previous studies [72].

Nevertheless, it is important to highlight that the information provided in the context of the ISA-Capital survey included data on other lifestyle characteristics (e.g., tobacco use and physical activity) that have been linked to alcohol abuse and other health behaviors missing from the present dataset [73], particularly among Brazilian adolescents [74,75]. In addition, evidence in the present study was obtained through statistical analyses performed on survey data that had been acquired via sampling procedures designed to ensure representativeness at population level, in addition to the use of empirical strategies based on robust estimation methods, thus ensuring minimization of potential errors.

Finally, the contributions of the study indicate that lifestyle and cardiometabolic risk factors represent determinants for impoverishment and may thus be important targets for health public policies aiming at promoting greater financial protection against OOP health expenditure in the Brazilian population. Public sector investment in preventive strategies at the primary healthcare level may be the key to reducing the rate of complex clinical conditions including obesity, CVD, HBP, and DM in São Paulo city. The findings of this study also suggest that additional investigation into determinants of lifestyle and cardiometabolic risk factors is required to provide further evidence for the design of public policies to foster access and adherence to preventive care in Brazil.

## 6. Conclusions

The evidence provided by the present study indicates the importance of targeting financial protection in relation to healthcare within the Brazilian two-tier health system in the context of the population of São Paulo city and may potentially contribute to enhancing universal health coverage through evidence-based policy design. Lifestyle and cardiometabolic risk factors related to physical activity, obesity, CVD, and DM comprise substantial determinants of impoverishment due to OOP health expenditure. The income-related inequalities in occurrence of OOP health expenditure and impoverishment present a regressive incidence, disproportionally concentrated among low-income households in São Paulo city, Brazil.

Public health policies directed towards financial protection of the population should focus on preventive strategies targeting modifiable risk factors, particularly interventions for maintenance of healthy body weight through promotion of active lifestyles. The incorporation of equity-oriented policies integrating intersectoral interventions directed at low- and middle-income households may ensure inclusive access and adherence to healthy lifestyles, especially in large urban centers in developing countries marked by socioeconomic inequalities, particularly in Latin America.

## Figures and Tables

**Figure 1 ijerph-21-01250-f001:**
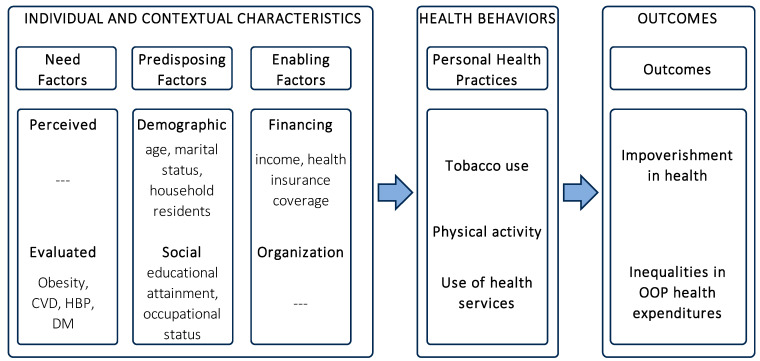
Diagram of the adaptation of Andersen’s Behavioral Model for Health Services Use [30].

**Table 1 ijerph-21-01250-t001:** Descriptive statistics of the study.

Variable	Categories	N	Weighted Mean	SD	Min	Max
**Outcomes**						
Impoverishment, IPL < 1.90	(0 = no/1 = yes)	5475	0.026	0.003	0	1
Impoverishment, IPL < 3.20	(0 = no/1 = yes)	5475	0.037	0.003	0	1
Impoverishment, IPL < 5.50	(0 = no/1 = yes)	5475	0.056	0.004	0	1
OOP health expenditure per capita	($PPP)	5475	128.68	320.19	0	6469.97
**Variables of interest**						
Tobacco use	(0 = no/1 = yes)	5475	0.460	0.010	0	1
Recommended PA level in leisure	(0 = no/1 = yes)	5475	0.373	0.010	0	1
Obesity	(0 = no/1 = yes)	5475	0.273	0.009	0	1
CVD	(0 = no/1 = yes)	5475	0.099	0.006	0	1
HBP	(0 = no/1 = yes)	5475	0.289	0.009	0	1
DM	(0 = no/1 = yes)	5475	0.099	0.006	0	1
**Control variables**						
Household with >4 residents	(0 = no/1 = yes)	5475	0.279	0.009	0	1
Presence of elderly	(0 = no/1 = yes)	5475	0.031	0.003	0	1
Head of family with higher education	(0 = no/1 = yes)	5475	0.240	0.009	0	1
Head of family employed	(0 = no/1 = yes)	5475	0.730	0.008	0	1
Head of family married	(0 = no/1 = yes)	5475	0.697	0.008	0	1
Household with high income	(0 = no/1 = yes)	5475	0.408	0.009	0	1
PHI coverage	(0 = no/1 = yes)	5475	0.349	0.009	0	1
Outpatient service use	(0 = no/1 = yes)	5475	0.749	0.007	0	1
Inpatient service use	(0 = no/1 = yes)	5475	0.148	0.007	0	1
Dental service use	(0 = no/1 = yes)	5475	0.432	0.010	0	1
Year	(2003; 2008; 2015)	5475			2003	2015

SD = standard deviation; Min = minimum; Max = maximum. IPL = international poverty line; CVD = cardiovascular diseases; HBP = high blood pressure; DM = diabetes mellitus; PA = physical activity; PHI = private health insurance; OOP = out-of-pocket.

**Table 2 ijerph-21-01250-t002:** Sociodemographic characteristics of the sample.

Variable		2003	2008	2015	2003–2015	*p*
N		1867	1565	2043	5475	
**Sociodemographic characteristics**						
Household with >4 residents	(1 = yes)	32.54 [29.29,35.98]	28.96 [25.64,32.51]	22.97 [20.51,25.63]	27.94 [26.18,29.78]	*
Presence of elderly	(1 = yes)	1.94 [1.35,2.78]	3.33 [2.60,4.26]	3.90 [2.98,5.09]	3.10 [2.63,3.65]	
Head of family with higher education	(1 = yes)	20.65 [17.83,23.77]	22.47 [19.28,26.01]	28.41 [25.97,30.99]	24.03 [22.37,25.76]	*
Head of family employed	(1 = yes)	72.54 [69.56,75.34]	73.32 [70.32,76.11]	73.24 [70.83,75.53]	73.05 [71.46,74.58]	
Head of family married	(1 = yes)	70.75 [67.6,73.71]	70.54 [67.4,73.49]	68.14 [65.65,70.52]	69.74 [68.08,71.35]	
Household with high income	(1 = yes)	40.19 [36.91,43.56]	39.94 [36.37,43.61]	42.17 [39.5,44.9]	40.82 [38.98,42.69]	

Data presented in weighted frequencies and 95% confidence intervals. *p*-values from χ^2^ test for comparison of survey years (2003 and 2015) estimate evolution. * *p* < 0.05.

**Table 3 ijerph-21-01250-t003:** Lifestyle, cardiometabolic, and health demand characteristics of the sample.

Variable		2003	2008	2015	2003–2015	*p*
N		1867	1565	2043	5475	
**Lifestyle characteristics**						
Tobacco use	(1 = yes)	41.74 [38.37,45.18]	53.31 [49.79,56.8]	43.24 [40.55,45.98]	46.04 [44.18,47.92]	*
Recommended PA level during leisure	(1 = yes)	30.76 [27.53,34.2]	42.03 [38.47,45.68]	35.75 [33.06,38.53]	36.22 [34.37,38.11]	*
**Health characteristics**						
Obesity	(1 = yes)	23.19 [20.35,26.28]	24.49 [21.42,27.86]	33.6 [31,36.3]	27.35 [25.68,29.09]	*
CVD	(1 = yes)	5.43 [3.88,7.55]	8.39 [6.74,10.39]	15.14 [13.12,17.4]	9.87 [8.79,11.08]	*
HBP	(1 = yes)	18.88 [16.26,21.82]	33.38 [30.06,36.86]	33.58 [31.04,36.22]	28.87 [27.2,30.61]	*
DM	(1 = yes)	6.03 [4.39,8.22]	10.04 [8.08,12.41]	13.21 [11.44,15.2]	9.91 [8.82,11.12]	*
**Healthcare characteristics**						
PHI coverage	(1 = yes)	36.33 [33.1,39.69]	36.17 [32.69,39.79]	32.36 [29.79,35.05]	34.85 [33.05,36.7]	
Physician visit	(1 = yes)	59.76 [56.55,62.9]	77.05 [74.42,79.48]	86.27 [84.7,87.69]	74.9 [73.43,76.32]	*
Hospitalization	(1 = yes)	10.09 [8.042,12.58]	15.91 [13.3,18.92]	17.91 [15.72,20.32]	14.79 [13.41,16.27]	*
Dentist visit	(1 = yes)	38.41 [35.14,41.78]	48.19 [44.62,51.77]	42.85 [40.14,45.61]	43.18 [41.33,45.06]	*

Data presented in weighted frequencies and 95% confidence intervals. CVD = cardiovascular disease; HBP = high blood pressure; DM = diabetes mellitus; PHI = private health insurance; PA = physical activity. *p*-values from χ^2^ test for comparison of survey years (2003 and 2015) estimate evolution. * *p* < 0.05.

**Table 4 ijerph-21-01250-t004:** Incidence of impoverishment due to OOP health expenditure.

Variable		2003	2008	2015	2003–2015	*p*
N		1867	1565	2043	5475	
**Impoverishment due to OOP health expenditure**						
IPL ≤ 1.90	(1 = yes)	3.21 [2.21,4.63]	2.28 [1.51,3.43]	2.26 [1.64,3.10]	2.56 [2.07,3.16]	
IPL ≤ 3.20	(1 = yes)	4.09 [3.13,5.33]	3.51 [2.47,4.98]	3.53 [2.70,4.62]	3.69 [3.11,4.38]	
IPL ≤ 5.50	(1 = yes)	7.09 [5.58,8.96]	5.01 [3.64,6.87]	5.14 [4.05,6.49]	5.64 [4.85,6.56]	
**Impoverishment according to income level**						
IPL ≤ 1.90—High income	(1 = yes)	0.50 [0.08,2.87]	1.14 [0.19,6.48]	0.99 [0.36,2.69]	0.80 [0.35,1.79]	
IPL ≤ 1.90—Low and middle income	(1 = yes)	5.3 [3.63,7.67]	3.25 [2.16,4.87]	3.40 [2.41,4.77]	3.93 [3.16,4.89]	
IPL ≤ 3.20—High income	(1 = yes)	0.61 [0.13,2.67]	1.58 [0.39,6.16]	1.11 [0.44,2.78]	0.97 [0.48,1.94]	
IPL ≤ 3.20—Low and middle income	(1 = yes)	7.13 [5.44,9.29]	5.37 [3.75,7.63]	5.66 [4.25,7.48]	5.99 [5.03,7.11]	
IPL ≤ 5.50—High income	(1 = yes)	0.67 [0.17,2.60]	1.58 [0.39,6.16]	1.79 [0.89,3.58]	1.26 [0.70,2.25]	
IPL ≤ 5.50—Low and middle income	(1 = yes)	14.81 [11.68,18.59]	8.47 [6.13,11.6]	8.17 [6.44,10.31]	9.96 [8.55,11.56]	*

Data presented in weighted frequencies and 95% confidence intervals. IPL = international poverty line. *p*-values from χ^2^ test for comparison of survey years (2003 and 2015) estimate evolution. * *p* < 0.05.

**Table 5 ijerph-21-01250-t005:** Inequity in out-of-pocket health expenditure.

Year	Concentration Index (CI)	Gini (G)	Kakwani Index (CI-G)
2003	0.467	0.661	−0.194
2008	0.447	0.578	−0.131
2015	0.423	0.471	−0.048
2003–2015	0.448	0.569	−0.121

CI for the distribution of OOP health expenditure in the sample ranked by household income per capita; Gini index for the distribution of the household income per capita.

**Table 6 ijerph-21-01250-t006:** Coefficients of predictors associated with OOP health expenditure.

Variable		Logistic			GLM			ME		
		β	SE	Sig	β	SE	Sig	dy/dx	SE	Sig
Household with >4 residents	(ref. = yes)	0.003	0.104		0.097	0.070		31.14	23.39	
Presence of elderly individuals	(ref. = yes)	0.543	0.225	*	0.271	0.109	*	117.29	37.64	**
Head of family with higher education	(ref. = yes)	0.796	0.129	***	0.608	0.073	***	239.57	26.48	***
Head of family employed	(ref. = yes)	0.045	0.096		−0.320	0.068	***	−100.11	22.88	***
Head of family married	(ref. = yes)	0.406	0.092	***	0.408	0.063	***	153.52	22.16	***
Household with high income	(ref. = yes)	0.832	0.103	***	0.814	0.067	***	307.77	27.25	***
Tobacco use	(ref. = yes)	−0.158	0.087		0.054	0.061		8.51	20.31	
Recommended PA level during leisure	(ref. = yes)	−0.019	0.055		−0.122	0.034	***	−40.18	11.46	***
Obesity	(ref. = yes)	−0.106	0.102		0.183	0.063	**	52.75	20.90	*
CVD	(ref. = yes)	0.703	0.179	***	0.221	0.098	*	110.47	33.26	**
HBP	(ref. = yes)	0.294	0.112	**	0.065	0.073		37.47	24.39	
DM	(ref. = yes)	0.309	0.158		0.203	0.096	*	82.39	31.95	*
N		5475			5475			5475		

GLM = generalized linear model; ME = marginal effect; β = regression coefficient; dy/dx = marginal effects; SE = robust standard error; CVD = cardiovascular disease; HBP = high blood pressure; DM = diabetes mellitus; PA = physical activity. Models included control variables for years of the survey. Sig. = * *p* < 0.05; ** *p* < 0.01; *** *p* < 0.001.

**Table 7 ijerph-21-01250-t007:** Coefficients of models for occurrence of impoverishment due to OOP health expenditure.

Variable	IPL ≤ 1.90			IPL ≤ 3.20			IPL ≤ 5.50		
	OR	SE	Sig	OR	SE	Sig	OR	SE	Sig
Household with >4 residents	1.813	0.404	**	1.491	0.294	*	0.973	0.200	
Presence of elderly	0.969	0.324		1.346	0.523		1.291	0.472	
Head of family with higher education	1.052	0.441		0.966	0.346		0.945	0.286	
Head of family employed	0.505	0.122	**	0.564	0.115	**	0.812	0.150	
Head of family married	0.972	0.224		1.261	0.256		1.308	0.236	
Household with high income	0.359	0.165	*	0.312	0.128	**	0.252	0.101	**
Tobacco use	1.689	0.411	*	1.128	0.217		1.114	0.198	
Recommended PA level during leisure	0.868	0.117		0.766	0.081	*	0.789	0.085	*
Obesity	0.823	0.216		1.588	0.310	*	1.633	0.316	*
CVD	2.268	0.737	*	1.967	0.492	**	1.936	0.508	*
HBP	0.947	0.320		1.185	0.272		1.324	0.284	
DM	2.506	0.797	**	1.161	0.300		1.342	0.306	
PHI coverage	1.258	0.295		0.639	0.134	*	0.629	0.122	*
Household with physician visit	1.174	0.254		1.269	0.240		1.045	0.194	
Household with hospitalization	1.005	0.302		1.705	0.410	*	0.853	0.229	
Household with dentist visit	1.604	0.333	*	1.365	0.240		1.828	0.298	***
N	5475			5475			5475		

OR = odds ratio; SE = robust standard error; IPL = international poverty line; CVD = cardiovascular disease; HBP = high blood pressure; DM = diabetes mellitus; PHI = private health insurance; PA = physical activity. Models included control variables for years of the survey. Sig. = * *p* < 0.05; ** *p* < 0.01; *** *p* < 0.001.

## Data Availability

The datasets generated and/or analyzed in the current study are available from the corresponding author upon reasonable request.

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
