# Peer review of "Lifestyle and Cardiometabolic Risk Factors Associated with Impoverishment Due to Out-of-Pocket Health Expenditure in São Paulo City, Brazil"

_ijerph, 2024, doi:10.3390/ijerph21091250_

Round 1

Reviewer 1 Report

Comments and Suggestions for Authors

Major concerns.

1.  The Andersen's model should be introduced in a separate section “Theoretical framework” after the Introduction (not in a sub-section “Materials and Methods”). Its assumptions, strengths, and limitations should be properly presented, grounding the current study better to the previous literature. The manuscript would benefit from development a few non-trivial theory-based hypotheses with their further testing within the statistical analysis.

2. Importantly, similar calculations with respect to Brazilian national poverty line should be provided for years 2003, 2008, and 2015. Alternatively, a clarification on correspondence of the national poverty thresholds to their international analogues for respective years should be added to sub-section "Outcome variable".

3. The current results are based on data that miss some important predictors, which can affect impoverishment, such as, for example, alcohol consumption, etc. Therefore, the results might be subject to omitted variable bias. All potentially relevant variables should be included in the statistical analysis. In case of their absence in the survey data, it should be clearly stated among major data-driven limitations of the study.

4. It is necessary to clarify to what extent the policy recommendations are applicable to wider population, for example, of large Brazilian cities, Brazil in general, large Latin American cities, etc.

5. Consider re-arranging the Discussion by highlighting the novel findings (from additional analyses, if necessary) in the beginning of the section. The current version simply “present complementary evidence” (lines 384-385).

Minor remarks.

1. Consider global replacing “factors [associated with impoverishment]” by “predictors [associated with impoverishment]”. Terminologically, this would be correct as factor analysis is a different method.

2. Revise the wording of line 113 because the use of word “interviewed” is inappropriate for survey respondents.

3. In the heading of the 4th column of Table 1, replace “nu” by “Weighted mean” (and remove the respective note below the table).

4. In lines 355-357, replace informal tilde signs (~) by formal words, which you have in mind.

Reviewer 2 Report

Comments and Suggestions for Authors

Review of Lifestyle and cardiometabolic risk factors associated with impoverishment due to out-of-pocket health expenditures in São 3 Paulo city, Brazil

For International Journal of Environmental Research and Public Health

As included in the manuscript, the study investigated trends and factors associated with impoverishment due to health expenditures in the population of São Paulo city, Brazil, between 2003 and 2015. Household data from the São Paulo Health Survey (n=5,475) were used to estimate impoverishment linked to OOP health expenses in São Paulo city, Brazil, using the three thresholds of International Poverty Lines (IPL) defined by the World Bank at 1.90, 17 3.20, and 5.50 dollars per capita per day in purchasing power parity (PPP) in 2011.

Overarching grammar editing required with particular attention to the use of articles and prepositions

Specific Comments—

More information about the survey—2003, 2008 and 2015 focus on Nutrition—do any more recent surveys do so? 

Expenditures adjusted for inflation across the years of the study—to 2015, were the IPLs also adjusted to 2015 values?

Lay out the Andersen’s Behavioral Model for Health Services 123 Use in diagram that shows how the variables relate to this model

Self -reported outcome variables

In tables reporting share of sample at different IPLS—at=re the IPL is inclusive? Ie does IPL5.50 include IPL1.90 and IPL3.20?

Why are all covariates indictors?  Could they be more detailed—is income continuous? Is health service use only reported as binary? Are there more categories for marital status and education for example.

What cutoffs identify a household as high income, higher education?

Comments on the Quality of English Language

Overarching grammar editing required with particular attention to the use of articles and prepositions

Round 2

Reviewer 1 Report

Comments and Suggestions for Authors

Dear authors, the revised version of manuscript is improved. However, the following minor issues should be addressed before publication.

1. Consider naming Table 1 as “Descriptive statistics”

2. Consier removing “Sao Paulo city, Brazil, 2003-2015” from the titles of your Tables.

3. In Line 388, provide a reference to a study, in which Kakwani index was introduced.

4. Doublecheck the calculations of Kakwani indices for years 2008 and 2003-2015.

Comments on the Quality of English Language

1. In line 451, fix a typo “representsing” (should be “represents”).

2. Revise the wording of line 183 because the use of word “interviewed” is inappropriate for survey respondents.

3. Revise the first sentence of the Discussion as its current version is confusing. It would be useful to split it into two sentences.

4. Again, let me draw you attention that the use of the term “factors” is often incorrect and should be replace by “predictors”.
